# Correlation and Correspondence between Skeletal Maturation Indicators in Hand-Wrist and Cervical Vertebra Analyses and Skeletal Maturity Score in Korean Adolescents

**DOI:** 10.3390/children8100910

**Published:** 2021-10-13

**Authors:** Ji Yoon Jeon, Cheol-Soon Kim, Jung-Suk Kim, Sung-Hwan Choi

**Affiliations:** 1Department of Orthodontics, Institute of Craniofacial Deformity, Yonsei University College of Dentistry, Seoul 03722, Korea; jiyoonjeon@hanmail.net; 2Private Clinic, Seongnam-si 13615, Korea; warmday@hanmail.net

**Keywords:** skeletal maturity, skeletal maturation indicators, cervical vertebrae maturation indicators, skeletal maturity score, adolescents

## Abstract

This retrospective observational study aimed to examine the correlation and correspondence between skeletal maturation indicators (SMI), cervical vertebral maturation indicators (CVMI), and radius-ulna-short bones (RUS) skeletal maturity scores in Korean adolescents, and to determine whether easily obtainable SMI or CVMI can replace the RUS skeletal maturity score. A total of 1017 participants were included with both hand-wrist radiograph and lateral cephalogram acquired concurrently. From the lateral cephalogram, CVMI was determined; through the hand-wrist radiograph, SMI was categorized, and the RUS skeletal maturity score was evaluated as well. Associations were examined using the Mann–Whitney U test, Spearman’s rank-order correlation analysis, and multiple correspondence analysis. There was no statistically significant difference in chronological age between males and females; however, the SMI, CVMI, and RUS skeletal maturity scores were significantly higher in females. The SMI, CVMI, and RUS skeletal maturity scores showed a statistically significant strong degree of both positive correlation and correspondence. However, a precisely corresponding RUS skeletal maturity score was difficult to obtain for a specific CVMI and SMI stage, implying the absence of a quantitative correlation. In conclusion, detailed evaluation should be conducted using the RUS skeletal maturity score, preferably in cases that require bone age determination or residual growth estimation.

## 1. Introduction

Orthodontics requires accurate predictions of the growth of adolescent patients to determine the timing of treatment. In this context, to account for individual diversity, the developmental age is used rather than chronological age as the indicator of interest [1]. Skeletal maturity evaluation methods are used in clinical practice as they are associated with changes in height and are relatively easy to perform through radiograph analysis [2]. Among them, Fishman’s skeletal maturity indicators (SMI) are categorized into 11 stages, based on the bone maturity observed in hand-wrist radiographs [3]. Cervical vertebrae maturation indicators (CVMI) evaluate the cervical vertebrae maturity based on a lateral cephalogram, which allows orthodontists to estimate the approximate period of entry into puberty, maximum growth, and growth termination.

In other contexts, such as in pediatrics or growth clinics, methodologies other than SMI are mainly used to evaluate skeletal maturity from hand-wrist radiographs [4,5], including the Tanner-Whitehouse 3 method (TW3 method), which determines the radius, ulna, short bones (RUS) skeletal maturity score (RUS skeletal maturity score) with weighting and scoring the maturity of each bone in the hands and wrists [5].

The RUS skeletal maturity score allows the calculation of bone age using a conversion table of the TW3 method and also the prediction of height at growth completion [5]. Consequently, it is used in many growth clinics. In contrast, orthodontic growth evaluation methods require additional radiographs alongside the lateral cephalograms obtained at the time of orthodontic diagnosis for the SMI evaluation. Furthermore, SMI does not allow the identification of the specific bone age, or to predict the amount of residual growth, thus limiting the accuracy of growth evaluation.

In practice, many patients and guardians who visit the clinic for orthodontic treatment often inquire after the expected height at the time of growth completion or the residual amount of statural growth. Provided that inferring the RUS skeletal maturity score from the SMI and CVMI is possible based on a correspondence between these factors, orthodontists may be able to report more instructive statural growth-related information to patients and guardians. However, studies into the correlation and correspondence between the SMI, CVMI, and RUS skeletal maturity scores in Korean adolescents are limited.

Thus, this retrospective observational study aimed to examine the correlation and correspondence between the SMI, CVMI, and RUS skeletal maturity scores in Korean adolescents, and to determine whether easily obtainable SMI or CVMI can replace the RUS skeletal maturity score with high confidence. The null hypothesis was that RUS skeletal maturity score can be logically deduced from the SMI or CVMI.

## 2. Materials and Methods

### 2.1. Samples

This study included 1017 (403 males and 614 females) participants with a mean age of 11.9 ± 2.5 (range, 4.9–18.8; median, 12.1) years who visited private clinics and for whom both hand-wrist radiograph and lateral cephalogram examinations were concurrently performed between August 2019 and February 2021 (Table 1).

Exclusion criteria were as follows: (1) age of >19 years, (2) severe dentofacial anomalies such as a cleft lip or palate, (3) previous history of growth hormone therapy, (4) presence of chronic disease or history of medication, and (5) radiographs with quality precluding skeletal maturity evaluation.

The study protocol adhered to the tenets of the Declaration of Helsinki (2013) and was approved by the Institutional Review Board of Yonsei Dental Hospital (IRB No. 2-2021-0071). The ethical approval and informed consent were waived in view of the retrospective nature of the study; all performed procedures were part of routine care. 

The effect size for the analyses was determined via G*Power ver. 3.1.9.2 (Dusseldorf University, Dusseldorf, Germany) with *p*-values of <0.05, a power of 80%, and an effect size of 0.1; the minimum sample size was 612 to enable the assessment.

### 2.2. Radiograph Analysis

Both lateral cephalograms and hand-wrist radiographs were obtained with a multipurpose imaging system (PaX-i3D Smart; Vatech Co., Gyeonggi-Do, Korea). Hand-wrist radiographs were obtained following the manufacturer’s instructions, placing a patient’s left hand on the carpus plate without displaying the nasal positioner. The radius and ulna of the wrist were then exposed, and exposure factors (exposure of 0.2 s, 10 mA, and 80 kVp for lateral cephalogram; exposure of 0.2 s, 5 mA, and 60 kVp for hand-wrist) were defined.

CVMI was determined using the Hassel’s classification method by evaluating the contour of the lower border of the second, third, and fourth cervical vertebral bodies and the height of the cervical vertebrae in the lateral cephalogram and classifying them from stage 1 to stage 6 [6].

SMI was categorized from stage 1 to stage 11 according to Fishman’s SMI method by evaluating the thumb, third finger, fifth finger, and radius through a hand-wrist radiograph [3]. To determine the RUS skeletal maturity score, each skeletal maturity of 13 bones of the hand and wrist was evaluated and mathematically converted into corresponding scores using the rating system of the TW3 method, followed by a summation of the total [5] (Figure 1).

### 2.3. Reproducibility

All assessments of skeletal maturity were performed by one orthodontist with 23 years of clinical experience. Intra-observer reproducibility was investigated by comparing the values of the original with those of re-evaluated variables 2 weeks after the first measurement in a randomly chosen set of 50 subjects. The intraclass correlation coefficient was of >0.97 for all evaluation values.

### 2.4. Statistical Analysis

All statistical analyses were performed in IBM SPSS Statistics for Windows (version 20.0; IBM Corp., Armonk, NY, USA), and the results were considered statistically significant at *p*-values of <0.05. The Shapiro–Wilk test was used to verify the normality of data distribution in the RUS skeletal maturity score; the data were non-normally distributed and thus a nonparametric test was used.

The Mann–Whitney U test was used to compare chronological age, the SMI, CVMI, and RUS skeletal maturity scores between males and females. Spearman’s rank-order correlation analysis was used to evaluate correlations between the SMI, CVMI, and RUS skeletal maturity score (*r* of <0.4, weak correlation; 0.4 < *r* < 0.6, moderate correlation; *r* of >0.6, strong correlation) [7]. In addition, multiple correspondence analysis was conducted between the SMI, CVMI, and RUS skeletal maturity scores to examine patterns and to determine the corresponding values between the variables. The strongest corresponding RUS skeletal maturity score for each SMI and CVMI stage was determined by calculating the distance between two points, using the principal coordinates of the first and second dimensions, and finding the ones at the shortest distance.

## 3. Results

The descriptive statistics of participants’ SMI, CVMI, and RUS skeletal maturity scores are presented in Table 1. There was no statistically significant difference in chronological age between males and females (*p* = 0.053); however, the SMI, CVMI, and RUS skeletal maturity scores were significantly higher in females than in males (*p* < 0.001). 

Spearman’s rank-order correlation analysis showed a statistically significant strong degree of positive correlation between the SMI, CVMI, and RUS skeletal maturity scores in the overall sample and in both male and female subgroups (Table 2).

The two-dimension multiple correspondence analysis in all subjects revealed that the first and second dimensions had eigenvalues of 2.956 and 2.793, inertia of 0.985 and 0.931, and Cronbach’s alpha of 0.993 and 0.963, respectively. Regarding sex, the corresponding values presented in males were 2.977 and 2.831, 0.992 and 0.944, 0.996 and 0.970, respectively. The corresponding values presented in females were 2.956 and 2.844, 0.985 and 0.948, 0.993 and 0.973, respectively. Since all measurements conducted in this study were related to the skeletal maturity index, Cronbach’s alpha was very high [8,9]. Discrimination measures (Table 3 and Figure 2) and joint plots of category points were obtained (Figure 3) in all subjects and both sexes. 

The observed correspondence was CVMI 1-SMI 1, CVMI 4-SMI 7, CVMI 5-SMI 9, and CVMI 6-SMI 11 in males, and CVMI 1–SMI 1, CVMI 2–SMI 4, CVMI 3–SMI 6, CVMI 4–SMI 7, CVMI 5–SMI 9, and CVMI 6–SMI 11 in females (Figure 3). The corresponding values of the RUS skeletal maturity score for each CVMI and SMI stage are shown in Table 4.

## 4. Discussion

This study (*n* = 1017) aimed to determine the correlation and correspondence between the SMI, CVMI, and RUS skeletal maturity scores obtained from hand-wrist radiographs and lateral cephalograms.

Patient’s skeletal maturity may determine the treatment strategy in orthodontics, which depends on the growth status and residual growth of the maxillofacial skeleton. Although information on chronological age and secondary sexual characteristics can be easily obtained using questionnaires, these methods have a limited capacity to accurately evaluate bone maturity. Therefore, previous studies aimed to evaluate skeletal maturity using radiographs such as SMI, CVMI, and dental age, all of which are associated with maxillofacial growth [10,11,12,13]. However, facial bone growth is influenced not only by general body growth, but also by neural growth [14]; consequently, orthodontists mainly choose SMI and CVMI, as it can be determined easily and intuitively at the chairside over the RUS skeletal maturity score, which is a more accurate, but relatively complex, method focused on general body growth assessment.

Fishman’s SMI, which evaluates 6 parts with 2–3 stages in the hand-wrist radiograph, is a much more simplified method compared to the TW3 method, which weights and evaluates 13 parts, respectively. Therefore, this method is commonly used at the chairside to determine the growth status of growing patients presenting at orthodontic clinics. This method is particularly useful for evaluating the pubertal growth spurt or growth termination with the emergence of the sesamoid bone or the fusion of the ulnar bone, which relatively limits disagreement between observers. However, an accurate assessment of bone maturity remains a challenge due to the likelihood of interobserver disagreement in evaluating the width and the capping of the phalanx at each finger. In addition, considering the fact that the development of the hand and wrist bones follows the Scammon’s general body growth curve [14], SMI has a restricted association, with other information related to growth; in contrast, the bone age and anticipated height can be possibly estimated from the RUS skeletal maturity score, which can be obtained from the same hand-wrist radiograph.

CVMI, which can be obtained by analyzing a lateral cephalogram, determines skeletal maturity as stage 1–6 by evaluating three cervical vertebrae in 2–3 stages, and may be used as an alternative to Fishman’s SMI, as the two measures have a high correlation [15,16]. The CVMI can reliably determine the pubertal growth spurt period [6,17,18,19,20], and help evaluate the skeletal maturity using hand-wrist radiographs [21]; it also minimizes radiation exposure by eliminating the need for additional hand-wrist radiographs. However, as this method involves only 6 stages of evaluation from the continuously changing development, it does not account for individual diversity, precluding the type of detailed evaluations associated with the analysis of the hand-wrist radiographs. In addition, poor interobserver reproducibility is likely in determining the shape of cervical vertebrae as trapezoidal, rectangular, or square shape.

In the present study, a statistically significant strong degree of positive correlations were observed between the SMI, CVMI, and RUS skeletal maturity scores; these findings are consistent with those of previous studies [22,23]. However, neither a single CVMI or SMI stage corresponded to a single RUS skeletal maturity score value, nor did an increase in SMI or CVMI necessarily correspond to a higher RUS skeletal maturity score (Table 4). Although the overall trend showed a positive correlation, it was often difficult to find a precisely corresponding RUS skeletal maturity score for a specific CVMI and SMI, and the range of values was wide in scope, including multiple candidate corresponding scores. This finding implies that it is difficult to find a quantitative correlation between the SMI, CVMI, and RUS skeletal maturity score, and that a direct evaluation of the RUS skeletal maturity score is required for further accurate evaluation of skeletal maturity and residual statural growth due to its irreplaceableness with CVMI and SMI.

In addition, to the best of our knowledge, this study is first to show a sex-based difference in the relationship between SMI and CVMI. In the observed correspondence between CVMI and SMI, CVMI 1–SMI 1, CVMI 4–SMI 7, CVMI 5–SMI 9, and CVMI 6–SMI 11 showed strong associations in males (Figure 3b), whereas CVMI 1–SMI 1, CVMI 2–SMI 4, CVMI 3–SMI 6, CVMI 4–SMI 7, CVMI 5–SMI 9, and CVMI 6–SMI 11 showed strong associations in females (Figure 3c). In contrast to the findings from previous studies and the present study findings for females, SMI values corresponding to, or highly associated with, CVMI 2 and 3 were difficult to identify in the present study male participants. This finding suggests that an accurate evaluation of growth status is unlikely when the diagnosis of males is based on the assumption that CVMI 2 corresponds to SMI 3 and 4 and that CVMI 3 corresponds to SMI 5 and 6; therefore, the use of the RUS skeletal maturity score remains recommended for a more detailed evaluation.

This study had some limitations. First, this was a retrospective clinical study. Second, despite its large sample size, the distribution of patients was somewhat unequal, particularly in terms of males. To overcome these limitations and to confirm the correlation between various skeletal maturity indices and other clinical variables, further case-controlled studies are required. 

## 5. Conclusions

The null hypothesis of this study is rejected. The SMI and RUS skeletal maturity scores obtained from hand-wrist radiographs and the CVMI obtained from lateral cephalograms had a statistically significant strong degree of positive correlations in both sexes, and all three indicators can be used effectively used in orthodontic clinics. However, because the simplified methodology of evaluating skeletal maturity in 6 or 11 stages has a limited accuracy, the precision of evaluation should be improved using the RUS skeletal maturity score, preferably in specific cases that require the bone age determination or residual growth estimation. Therefore, it is recommended that clinicians should acquire a hand-wrist radiograph and perform a detailed evaluation as circumstances demand.

## Figures and Tables

**Figure 1 children-08-00910-f001:**
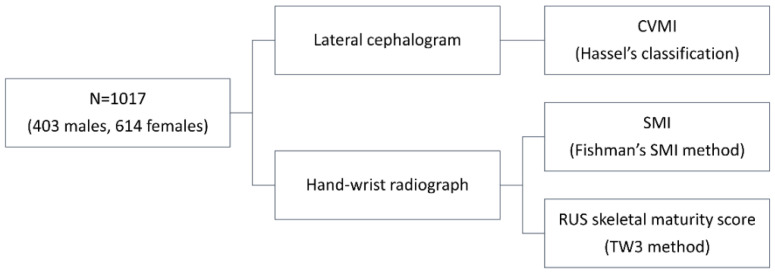
Schematic representation of the study protocol.

**Figure 2 children-08-00910-f002:**
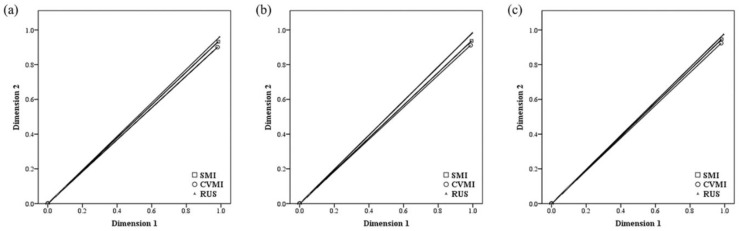
Multiple correspondence analysis dimension discrimination measures: (**a**) All subjects; (**b**) Males; (**c**) Females.

**Figure 3 children-08-00910-f003:**
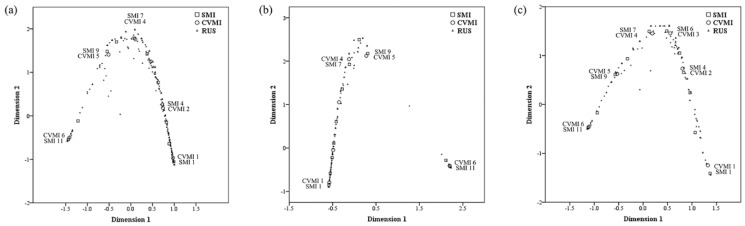
Multiple correspondence analysis joint plots of the SMI, CVMI, and RUS skeletal maturity scores: (**a**) All subjects; (**b**) Males; (**c**) Females.

**Table 1 children-08-00910-t001:** Participant characteristics.

	Total	Male	Female	*p* Value (Mann–Whitney U Test)
**Number of subjects**	1017	403	614	
**Chronological age (year)**	11.9 ± 2.5 ^a^ (range, 4.9–18.8; 95% CI (11.7–12.0); median, 12.1)	11.7 ± 2.6 ^a^ (range, 6.6–18.6; 95% CI (11.5–12.0); median, 11.8)	12.0 ± 2.4 ^a^ (range, 4.9–18.8; 95% CI (11.8–12.2); median, 12.3)	0.053
**SMI**	6.0 ± 3.9 ^a^ (range, 1.0–11.0; 95% CI (5.7–6.2); median, 6.0)	4.6 ± 3.5 ^a^ (range, 1.0–11.0; 95% CI (4.3–4.9); median, 3.0)	6.9 ± 3.8 ^a^ (range, 1.0–11.0; 95% CI (6.6–7.2); median, 7.0)	**<0.001 *****
**CVMI**	3.4 ± 2.0 ^a^ (range, 1.0–6.0; 95% CI (3.3–3.5); median, 3.0)	2.7 ± 1.8 ^a^ (range, 1.0–6.0; 95% CI (2.5–2.9); median, 2.0)	3.8 ± 2.0 ^a^ (range, 1.0–6.0; 95% CI (3.7–4.0); median, 4.0)	**<0.001 *****
**RUS skeletal maturity score**	613.8 ± 272.0 ^a^ (range, 167–1000; 95% CI (597.1–630.5); median, 592.0)	480.1 ± 256.6 ^a^ (range, 167–1000; 95% CI (455.0–505.3); median, 367.0)	701.6 ± 244.9 ^a^ (range, 209–1000; 95% CI (682.2–721.0); median, 726.5)	**<0.001 *****

CI, Confidence Interval. ^a^ Values are expressed as means ± standard deviations. Bold values indicate the results that are statistically significant (*** *p* < 0.001).

**Table 2 children-08-00910-t002:** Spearman’s rank correlation coefficient between SMI, CVMI, and RUS skeletal maturity scores.

	Total	Male	Female
**CVMI and SMI**	0.955 **(*p* < 0.001 ***)**	0.958 **(*p* < 0.001 ***)**	0.917 **(*p* < 0.001 ***)**
**SMI and RUS skeletal maturity score**	0.948 **(*p* < 0.001 ***)**	0.957 **(*p* < 0.001 ***)**	0.944 **(*p* < 0.001 ***)**
**CVMI and RUS skeletal maturity score**	0.921 **(*p* < 0.001 ***)**	0.935 **(*p* < 0.001 ***)**	0.890 **(*p* < 0.001 ***)**

Bold values indicate the results that are statistically significant (*** *p* < 0.001).

**Table 3 children-08-00910-t003:** Multiple correspondence analysis dimensions discrimination measures.

	Total			Male			Female		
	**Multiple** **Correspondence Analysis Dimensions**		**Mean**	**Multiple** **Correspondence Analysis Dimensions**		**Mean**	**Multiple** **Correspondence Analysis Dimensions**		**Mean**
	**1**	**2**		**1**	**2**		**1**	**2**	
**SMI**	0.984	0.933	0.958	0.993	0.937	0.965	0.982	0.946	0.964
**CVMI**	0.980	0.901	0.940	0.987	0.912	0.949	0.980	0.924	0.952
**RUS skeletal maturity score**	0.993	0.960	0.976	0.998	0.983	0.990	0.994	0.974	0.984
**Active total**	2.956	2.793	2.875	2.977	2.831	2.904	2.956	2.844	2.900
**% of ** **variance**	98.538	93.104	95.821	99.231	94.375	96.803	98.541	94.813	96.677

**Table 4 children-08-00910-t004:** RUS skeletal maturity scores associated with the SMI and CVMI.

Total (*n* = 1017)				Male (*n* = 403)				Female (*n* = 614)			
CVMI	RUS Skeletal Maturity Score	SMI	RUS Skeletal Maturity Score	CVMI	RUS Skeletal Maturity Score	SMI	RUS Skeletal Maturity Score	CVMI	RUS Skeletal Maturity Score	SMI	RUS Skeletal Maturity Score
1	328	1	277	1	277	1	249, 265, 266, 268, 273, 276, 286	1	411, 423, 427, 432	1	209, 243, 264, 265, 269, 273, 276, 277, 280, 283, 284, 286, 290, 291, 293, 298, 301, 302, 303, 305, 306, 308, 309, 311, 312, 319, 320, 321, 325, 328, 329, 336, 337, 338, 349, 354, 359, 361, 365, 366, 368, 373, 374, 378, 382, 385, 387, 390, 391, 393, 400, 403, 405, 409, 412, 422, 435, 436, 438, 447
2	467	2	314, 372, 424	2	318	2	326	2	455, 459, 474, 478, 505, 506, 521, 522, 557	2	404, 426
3	564	3	334, 374, 380	3	564	3	334, 380	3	565	3	464
4	635	4	295	4	785	4	241	4	635	4	473
5	779	5	534	5	689, 744, 768, 896	5	401	5	809	5	458
6	928	6	516	6	928	6	399	6	945	6	611
		7	724			7	586, 614			7	767
		8	745			8	745			8	796
		9	744, 759, 768, 774, 781, 822, 824, 830, 835			9	689, 744, 768, 896			9	809
		10	852			10	854			10	852
		11	1000			11	928			11	1000

## Data Availability

All relevant data are within the manuscript.

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
