# Peer review of "Correlation and Correspondence between Skeletal Maturation Indicators in Hand-Wrist and Cervical Vertebra Analyses and Skeletal Maturity Score in Korean Adolescents"

_children, 2021, doi:10.3390/children8100910_

Round 1

Reviewer 1 Report

Dear Authors 

The article is interesting and well written. 

It is a retrospective observational study to examine the correlation and correspondence between skeletal maturation indicators, cervical vertebral maturation indicators , and radius-ulna-short bones skeletal maturity scores in Korean adolescents.
It has also the aim to determine whether easily obtainable SMI or CVMI can replace the RUS skeletal maturity score. Authors conclude that all three indicators can be used effectively in orthodontic clinics.
In my opinion the sample is enough, the statistic is good and I have noticed only minor spelling errors.

I I have no suggestions to improve the article.

Best regards 

Author Response

We sincerely appreciate your kind comments.

Reviewer 2 Report

Well-conceived and interesting paper. I would not suggest any modifications, except for lines 42-43, where there seems to be a pasted text from the instructions for authors

Author Response

We sincerely appreciate your kind comments. The contents in the left side of the first page were a given format from the journal; Academic editor, Received/Accepted/Published date, Publisher’s Note, and Copyright. It is expected that proper modifications will be made by the publisher after revision.

Reviewer 3 Report

This retrospective observational study aimed to examine the correlation and correspondence between skeletal maturation indicators (SMI), cervical vertebral maturation indicators (CVMI), and radius-ulna-short bones (RUS) skeletal maturity scores in Korean adolescents, and to determine whether easily obtainable SMI or CVMI can replace the RUS skeletal maturity score.

The authors concluded that the SMI and RUS skeletal maturity scores obtained from hand-wrist radiographs and the CVMI obtained from lateral cephalograms had statistically significant strong degree of positive correlations in both sexes, and all three indicators can be used effectively in orthodontic clinics.

The manuscript is well written and present interesting findings. However, some modifications are needed:

  • I suggest adding a clear conclusion related to the aim at the end of the abstract.
  • L42-43 present the guidelines of the journal and should be removed.
  • Add some hypothesis at the end of the Introduction.
  • I suggest adding a schematic representation of the protocol
  • I suggest adding the effect size in the Statistical analysis and results.
  • I suggest adding some practical recommendations at the end of the conclusion

Author Response

Thank you for your constructive suggestion.

Comment 1: I suggest adding a clear conclusion related to the aim at the end of the abstract.

Our Response: Thank you for your constructive suggestion. The abstract has been modified based on your comments to make it easier to understand.

Revised text:

Abstract: This retrospective observational study aimed to examine the correlation and correspondence between skeletal maturation indicators (SMI), cervical vertebral maturation indicators (CVMI), and radius-ulna-short bones (RUS) skeletal maturity scores in Korean adolescents, and to determine whether easily obtainable SMI or CVMI can replace the RUS skeletal maturity score. A total of 1017 participants were included with both hand-wrist radiograph and lateral cephalogram acquired concurrently. From the lateral cephalogram, CVMI was determined; through the hand-wrist radiograph, SMI was categorized, and the RUS skeletal maturity score was evaluated as well. Associations were examined using the Mann-Whitney U test, Spearman’s rank-order correlation analysis, and multiple correspondence analysis. There was no statistically significant difference in chronological age between males and females; however, the SMI, CVMI, and RUS skeletal maturity scores were significantly higher in female. The SMI, CVMI, and RUS skeletal maturity scores showed statistically significant strong degree of both positive correlation and correspondence. However, a precisely corresponding RUS skeletal maturity score was difficult to obtain for a specific CVMI and SMI stage, implying the absence of a quantitative correlation. In conclusion, detailed evaluation should be conducted using the RUS skeletal maturity score, preferably in cases that require bone age determination or residual growth estimation.

Comment 2: L42-43 present the guidelines of the journal and should be removed.

Our Response: Thank you for your comment. The contents in the left side of the first page were a given format from the journal; Academic editor, Received/Accepted/Published date, Publisher’s Note, and Copyright. It is expected that proper modifications will be made by the publisher after revision.

Comment 3: Add some hypothesis at the end of the Introduction.

Our Response: Thank you for your valuable comment. The null hypothesis and relevant conclusion have been added based on your comments.

Revised text:

Introduction

Thus, this retrospective observational study aimed to examine the correlation and correspondence between the SMI, CVMI, and RUS skeletal maturity score in Korean adolescents, and to determine whether easily obtainable SMI or CVMI can replace the RUS skeletal maturity score with high confidence. The null hypothesis was that RUS skeletal maturity score can be logically deduced from the SMI or CVMI.

Conclusion

The null hypothesis of this study is rejected. The SMI and RUS skeletal maturity scores obtained from hand-wrist radiographs and the CVMI obtained from lateral cephalograms had statistically significant strong degree of positive correlations in both sexes, and all three indicators can be used effectively in orthodontic clinics. However, because the simplified methodology of evaluating skeletal maturity in 6 or 11 stages has limited accuracy, the precision of evaluation should be improved using the RUS skeletal maturity score, preferably in specific cases that require bone age determination or residual growth estimation. Therefore, it is recommended that clinicians should acquire hand-wrist radiograph and perform detailed evaluation as circumstances demand.

Comment 4: I suggest adding a schematic representation of the protocol.

Our Response: Thank you for your constructive suggestion. A figure representing schematic table of the protocol has been added based on your comments.

Revised text:  

Figure 1. Schematic representation of the study protocol

Comment 5: I suggest adding the effect size in the Statistical analysis and results.

Our Response: Thank you for your valuable comment. This relevant information has been added according to your comments.

Revised text:

2.1. Samples

This study included 1017 (403 males and 614 females) participants with the mean age of 11.9 ± 2.5 (range, 4.9-18.8; median, 12.1) years that visited private clinics and for whom both hand-wrist radiograph and lateral cephalogram examinations were concurrently performed between August 2019 and February 2021 (Table 1).

Exclusion criteria were as follows: (1) age of > 19 years, (2) severe dentofacial anomalies such as a cleft lip or palate, (3) previous history of growth hormone therapy, (4) presence of chronic disease or history of medication, and (5) radiographs with quality precluding skeletal maturity evaluation.

The study protocol adhered to the tenets of the Declaration of Helsinki (2013) and was approved by the Institutional Review Board of Yonsei Dental Hospital (IRB No. 2-2021-0071). The ethical approval and informed consent were waived in view of the retrospective nature of the study; all performed procedures were part of routine care.

The effect size for the analyses was determined via G*Power ver. 3.1.9.2 (Dusseldorf University, Dusseldorf, Germany) with p-values of < 0.05, a power of 80%, and an effect size of 0.1; the minimum sample size was 612 to enable the assessment.

Comment 6: I suggest adding some practical recommendations at the end of the conclusion

Our Response: Thank you for your comment. This relevant information has been added according to your comments.

Revised text:

The null hypothesis of this study is rejected. The SMI and RUS skeletal maturity scores obtained from hand-wrist radiographs and the CVMI obtained from lateral cephalograms had statistically significant strong degree of positive correlations in both sexes, and all three indicators can be used effectively in orthodontic clinics. However, because the simplified methodology of evaluating skeletal maturity in 6 or 11 stages has limited accuracy, the precision of evaluation should be improved using the RUS skeletal maturity score, preferably in specific cases that require bone age determination or residual growth estimation. Therefore, it is recommended that clinicians should acquire hand-wrist radiograph and perform detailed evaluation as circumstances demand.

Based on your valuable opinions, overall, the text has been revised to organize the core contents of previous research concisely. Thank you once again for your sincere comments and consideration.